# *Methanobrevibacter massiliense* and *Pyramidobacter piscolens* Co-Culture Illustrates Transkingdom Symbiosis

**DOI:** 10.3390/microorganisms12010215

**Published:** 2024-01-20

**Authors:** Virginie Pilliol, Mamadou Beye, Laureline Terlier, Julien Balmelle, Idir Kacel, Romain Lan, Gérard Aboudharam, Ghiles Grine, Elodie Terrer

**Affiliations:** 1IRD, AP-HM, MEPHI, IHU Méditerranée Infection, Aix-Marseille University, 13005 Marseille, France; drvirginiepilliol@gmail.com (V.P.); terlierlaureline@gmail.com (L.T.); julien.balmelle@etu.univ-amu.fr (J.B.); gerard.aboudharam@gmail.com (G.A.); 2Ecole de Médecine Dentaire, Aix-Marseille University, 13385 Marseille, France; romain.lan@univ-amu.fr; 3IHU Méditerranée Infection, 13005 Marseille, France; bemamadou@gmail.com (M.B.); idir.kacel@ap-hm.fr (I.K.); 4CNRS, EFS, ADES, Aix-Marseille University, 13385 Marseille, France

**Keywords:** *Archaea*, methanogen, *Methanobrevibacter massiliense*, *Synergistetes*, *Pyramidobacter piscolens*, hydrogen-free culture, dental pulp

## Abstract

Among oral microbiota methanogens, *Methanobrevibacter massiliense* (*M. massiliense*) has remained less studied than the well-characterised and cultivated methanogens *Methanobrevibacter oralis* and *Methanobrevibacter smithii*. *M. massiliense* has been associated with different oral pathologies and was co-isolated with the *Synergistetes* bacterium *Pyramidobacter piscolens* (*P. piscolens*) in one case of severe periodontitis. Here, reporting on two additional necrotic pulp cases yielded the opportunity to characterise two co-cultivated *M. massiliense* isolates, both with *P. piscolens*, as non-motile, 1–2-µm-long and 0.6–0.8-µm-wide Gram-positive coccobacilli which were autofluorescent at 420 nm. The two whole genome sequences featured a 31.3% GC content, gapless 1,834,388-base-pair chromosome exhibiting an 85.9% coding ratio, encoding a formate dehydrogenase promoting *M. massiliense* growth without hydrogen in GG medium. These data pave the way to understanding a symbiotic, transkingdom association with *P. piscolens* and its role in oral pathologies.

## 1. Introduction

Methanogenic archaea (methanogens) are constant inhabitants of the human digestive tract microbiota [1], are regularly detected in the oral cavity [2], neonatal meconium, gastric fluid [3,4,5] and faeces [6], and have been detected in ancient dental calculus samples collected as far back as the Neolithic period [7,8,9,10]. The association between methanogens and oral pathologies is ambiguous, as methanogens have been detected in saliva collected from apparently healthy individuals and, more specifically, from tobacco smokers [11] as well as in oral cavity samples collected from patients diagnosed with pericoronitis [12], gingivitis, periodontitis [13,14,15,16], peri-implantitis [17,18,19], endodontic infections [20,21,22,23,24], and dental abscesses [25]. It has been hypothesised that methanogens may contribute to microbial community dynamics, particularly through syntrophic interactions with bacteria. In the context of endodontic conditions, molecular methods have revealed the presence of methanogens, showing a higher prevalence in inflamed pulp [23] compared to necrotic pulp in both primary and secondary infections [20,21,22,24]. Notably, one study suggested a closer association between methanogens and symptomatic cases [21]. Furthermore, root canal treatment seems to be effective in eliminating methanogens [26]. Notably, none of these studies reported the successful cultivation of methanogens from pulp samples.

*Methanobrevibacter smithii* (*M. smithii*) and *Methanobrevibacter oralis* (*M. oralis*) are the two main methanogens detected by molecular methods and isolated by culture among the oral microbiota, with *M. oralis* being the most prevalent [13,14,15,16]. However, several studies have reported *M. oralis* like-species, leaving the diversity of the *Methanobrevibacter* genus in the oral microbiota [20,22,23,27] unknown.

Further studies revealed a third oral *Methanobrevibacter* phylotype [13,27], possibly interacting with a *Synergistetes* spp. in periodontitis and endodontic infections [27]. This was further detected as *Methanobrevibacter* sp. N13 in 45% of ancient dental calculus samples, while *M. oralis* was detected in only 20% of the same samples. Thus, *M. oralis* is less prevalent than *Methanobrevibacter* sp. N13 in historical populations than in modern populations [7]. *Methanobrevibacter* sp. N13 was later named *Methanobrevibacter massiliense* (*M. massiliense*) after it was co-isolated with the *Synergistetes Pyramidobacter piscolens* (*P. piscolens*) during an 18-month culture of a leftover dental plaque sample collected from a patient diagnosed with severe periodontitis [28]. Additionally, *M. massiliense* has been identified in Malian patients with oral pathologies [29] in cases of peri-implantitis [17], and in a challenging sinusitis case, where it was the sole microorganism identified [30]. These instances collectively underscore its potential relevance in various human pathological conditions. *M. massiliense*, however, remains an uncharacterised methanogen that may have implications in oral pathologies.

In this study, our aim was first to recover then to cultivate and isolate *M. massiliense* in a novel oral condition, endodontic infections. This provided an opportunity for comprehensive characterisation, including genomic and microscopic features, contributing to a more thorough understanding of this challenging methanogen. Additionally, we sought to assess the prevalence of *M. massiliense* compared to other *Methanobrevibacter* species and to explore potential associations with putative partners, notably among *Synergistetes* spp.

## 2. Materials and Methods

### 2.1. Ethical and Regulatory Statement

In this study, teeth were extracted as part of routine clinical practice. No teeth were specifically extracted for the present work. The extracted teeth were leftover clinical material, classified as “waste from healthcare activities”, in accordance with the French Public Health Code (Article L1211-2). Investigators orally verified that no patient objected to the use of leftover clinical samples, including extracted teeth, for research investigations. To ensure patient confidentiality and privacy, strict anonymisation procedures were implemented throughout the study so that any personal identifying information was removed from the extracted teeth, which were assigned a unique identification code. Teeth were extemporaneously examined for cavities and pulp vitality. The research team had no access to any patient-related data or information that could potentially compromise the anonymity of involved individuals. Accordingly, this study was approved by the Institut Méditerranée Infection (IHU) Ethics Committee No. 2022-14.

### 2.2. Sample Collection

A total of 32 molars and premolars were collected from 32 patients as part of routine practice by dentists and dental students under the supervision of a graduate dentist at the Pôle Odontologie Timone (Assistance Publique des Hôpitaux de Marseille) between September 2022 and February 2023. Twenty-two teeth had to be extracted for carious reasons and ten teeth were extracted as wisdom teeth. None were in the state of root debris or were fractured during the extraction (Appendix A). Each tooth was stored in a 50 mL Falcon tube (Thermo Fisher Scientific, Waltham, MA, USA) containing 10 mL of Dulbecco’s Phosphate Buffered Saline (DPBS) 1× water (Thermo Fischer Scientific, Waltham, MA, USA) at 4 °C for a maximum of 24 h. Manipulations were carried out under anaerobiosis in an anaerobic chamber to avoid exposing the dental pulp to oxygen (Don Whitley, Bingley, UK). Each tooth was immersed in a 10 mL 20% chlorhexidine di-gluconate solution (Gilbert Healthcare, Hérouville-Saint-Clair, France) for 15 min, rinsed with DNase/RNase-free distilled water (Thermo Fischer Scientific, Waltham, MA, USA), re-immersed in a 10 mL 70% alcohol solution (Gilbert Healthcare, Hérouville-Saint-Clair, France) for one minute, rinsed with DNase/RNase-free distilled water, and cleaned with sterile gauze soaked with 3% sodium hypochlorite and then rinsed again before being wiped with a new sterile gauze. Each tooth was then partially and longitudinally sectioned using a sterile diamond disk and fractured using a sterile syndesmotome (Hu-Friedy, Frankfurt am Main, Germany), and the dental pulp was extirpated using a sterile excavator (Hu-Friedy, Frankfurt am Main, Germany), one half being put in an Eppendorf tube containing 200 µL of DPBS 1× for molecular analysis and one half for methanogen isolation by culture.

### 2.3. Methanogen DNA Detection

DNA was directly extracted from the dental pulp with 200 µL of G2 buffer added, as previously described [31]. The EZ1 DNA Tissue Kit (Qiagen, Courtaboeuf, France) was used after sample pretreatment, consisting of overnight incubation at 56 °C with 20 µL proteinase K (Qiagen, Courtaboeuf, France), the addition of glass powder, incubation for 20 min at 100 °C, vortexing for 90 s at 6.5 m/s with FastPrep (MP Biomedical Europe, Illkirch, France) and, finally, centrifugation for five minutes at 17,000× *g*. Methanogen was first detected by real-time PCR (RT-PCR) based on the methanogen 16S rRNA gene, as previously described [32]. PCR-positive samples were sequenced by Sanger sequencing of the partial archaeal 16S rRNA gene (primers: SDArch0333aS15, 5′-TCCAGGCCCTACGGG-3′ and SDArch0958aA19, 5′-YCCGGCGTTGAMTCCAATT-3′), as previously described [20]. Generated sequences were assembled using the ChromasPro software (version 1.34), then aligned against the NCBI GenBank database using the Blast platform (https://blast.ncbi.nlm.nih.gov/Blast.cgi, accessed on 1 March 2023). Only *M. massiliense*-positive samples were selected for further culture and isolation attempts. The presence of *M. massiliense* in the corresponding culture was also confirmed by the same methods.

### 2.4. Culturing M. massiliense

Dental pulps which tested positive by PCR sequencing for *M. massiliense* were cultured in a Hungate tube containing 5 mL of SAB medium [33] consisting of 100 mg/L amoxicillin, 100 mg/L vancomycin, 100 mg/L imipenem, 50 mg/L daptomycin and 50 mg/L amphotericin B (Sigma Aldrich, Saint Quentin Fallavier, France), in a 5% hydrogen, 10% CO_2_, 85% nitrogen atmosphere at 37 °C for 15 days. Culture was monitored by methane detection with gas chromatography using a Clarus 580 chromatograph (Perkin Elmer, Waltham, MA, USA), as previously described [33]. *M. massiliense*-positive cultures were subcultured using two conditions: in one condition, 200 µL was inoculated in a Hungate tube containing 5 mL of SAB medium incubated under a 5% hydrogen, 10% CO_2_, 85% nitrogen atmosphere, and in the other condition, 200 µL was inoculated in liquid GG medium under a nitrogen atmosphere [34]. Both tubes were incubated at 37 °C for 15 days. Methane was detected at days 5, 10 and 15 as described above, and cultures were finally established in GG medium, as above. In parallel, 200 µL of PBS 1× water was inoculated under the same conditions as the negative controls. To identify bacteria co-culturing with *M. massiliense*, 100 µL of the culture was inoculated on a Columbia agar 5% sheep blood (COS) medium plate (bioMérieux, Marcy-Étoile, France), incubated under an anaerobic atmosphere using GasPak (Becton, Dickinson and Company, Franklin Lakes, NJ, USA) at 37 °C for 15 days. At 3-, 5-, 7-, 10- and 15-day time points, cultures were observed for colonies, which were subsequently cultured and identified using MALDI-TOF mass spectrometry.

### 2.5. Culture Microscopy

*M. massiliense*-*P. piscolens* co-cultures were examined for autofluorescence under a confocal epifluorescence microscope, LSM 900 (Carl Zeiss Microscopy GmbH, Jena, Germany), at the 63 × objective with immersion oil, as previously described [35]. Slides were prepared with 10 µL volume deposit and sealed under an anaerobic atmosphere. For scanning electron microscopy, 100 µL of each culture was added to 100 µL of 4× glutaraldehyde and stirred for 30 min at room temperature. Then, 20 µL of 10% phosphotungstic acid was added, stirred again for five minutes, and finally was cytocentrifugated for seven minutes at 800 rpm onto a glass slide (Cytospin^®^, Thermo Fischer Scientific, Waltham, MA, USA). The slides underwent analysis using the TM4000 plus Scanning Electron Microscope (Hitachi, Tokyo, Japan). Micrographs were obtained at high magnifications, ranging from ×1000 to ×2500, employing an accelerating voltage of 10 kV and using the Backscatter Electron (BSE) detector in a high vacuum environment. As a comparison control, *P. piscolens* strains P2428, Q3403, Q3381, Q8451 and Q8452, from both liquid and solid cultures, were microscopically observed, as described above. Moreover, fluorescent in situ hybridization (FISH) was performed directly on pulp sample No. 74 using the archaea-specific Arch915 probe Alexa 488 (5′-GTGCTCCCCCGCCAATTCCT-3′) [36] and the partial 16S rRNA bacterial domain specific EUB 338 probe Alexa 647 (5′-GCTGCCTCCCGTAGGAGT-3′) [37], as previously described [11]. Negative controls consisted of the same manipulation on methanogen 16S rRNA gene PCR-negative dental pulp and on DPBS water 1×. The universal DNA stain DAPI was also added. Slides were also observed with a confocal epifluorescence microscope LSM 900 (Carl Zeiss Microscopy GmbH, Oberkochen, Germany) at appropriate wavelengths.

### 2.6. MALDI-TOF Mass Spectrometry Culture

One millilitre of *M. massiliense* co-cultures Q8282 and Q8283 and 1 mL of the five *P. piscolens* strains (P2428, Q3381, Q3403, Q8451 and Q8452, Collection Souches Unité des Rickettsies, CSUR, WDCM 875, Marseille, France), cultured in GG medium, as well as one 5 µL loop resuspended in 1 mL of DPBS 1× water (Thermo Fischer Scientific, Waltham, MA, USA) of the five *P. piscolens* strains (P2428, Q3381, Q3403, Q8451 and Q8452), cultured in COS medium, were collected for protein extraction, as previously described [38]. Two deposits were made for each sample and the protein extract of *Escherichia coli* (*E. coli*) DH5α™ (Brüker Daltonics, Bremen, Germany) was used as a positive control and free matrix as a negative control. Peptide profiles were obtained using an Autoflex II mass spectrometer with a 337 nm nitrogen laser (Brüker Daltonics, Bremen, Germany). The AutoXecute software acquisition control tool (Flex control 3.0, Brüker Daltonics, Bremen, Germany) was applied for automated data acquisition and spectral comparisons were performed using FlexAnalysis 2.4 software (version 2.4, Brüker Daltonics, Bremen, Germany).

### 2.7. P. piscolens Antibiotic Susceptibility Testing and M. massiliense Isolation Attempts

*P. piscolens* strains Q8451, Q8452, P2428, Q3403 and Q3381 were tested for tetracycline susceptibility using the tetracycline E-test (bioMérieux, Marcy-Étoile, France). Tetracycline was selected with a view to isolating *M. massiliense* in pure culture from *P. piscolens* co-culture, as methanogens are resistant to tetracycline (MIC > 100 mg/L) [39], whereas *P. piscolens* is susceptible to tetracycline [40]. In detail, *P. piscolens* suspensions (1 McFarland) were subcultured on one COS plate without antibiotic as a growth positive control and one COS plate with tetracycline E-test. A third COS plate was mock-inoculated with 100 µL of DBS 1× water as a negative control. Plates were incubated under an anaerobic atmosphere using the GasPak anaerobic generator (Becton, Dickinson and Company, Franklin Lakes, NJ, USA) for five days at 37 °C. Furthermore, each *P. piscolens* strain was subcultured (in duplicate) in one Hungate tube containing 5 mL of GG medium under nitrogen atmosphere in addition to 50 mg tetracycline and another containing 100 mg of tetracycline. One additional Hungate tube, mock-inoculated with 100 µL of DBS 1× water, was used as a negative control. Tubes were incubated for five days at 37 °C. In an attempt to isolate *M. massiliense* in pure culture, the *M. massiliense*-*P. piscolens* co-culture Q8282 was subcultured (in duplicate) in five distinct conditions: 100 µL of a growing culture was inoculated in GG medium as a positive control; in GG medium supplemented with 100 mg tetracycline; in GG medium supplemented with 100 mg tetracycline and 3 g/L of acetate; in GG medium supplemented with 100 mg tetracycline with volatile fatty acids solution [33]; and in GG medium supplemented with 100 mg tetracycline with 3 g/L of acetate and volatile fatty acids solution. All tubes were incubated for three weeks at 37 °C. Negative controls encompassed the same five conditions, but with the inoculation of 100 µL of DPBS 1× water.

### 2.8. Whole Genome Sequencing and Analysis

DNA was extracted from 200 µL of each co-culture using the EZ1 DNA Tissue Kit (Qiagen, Courtaboeuf, France), as previously described [41]. The mixture was incubated overnight at 56 °C with 20 µL of proteinase K (Qiagen), followed by the addition of glass powder and incubation for 20 min at 100 °C. The mixture was then immediately vortexed for 90 s at 6.5 m/s using the FastPrep instrument (MP Biomedical Europe, Illkirch, France). Subsequently, the mixture was centrifuged for five minutes at 17,000× *g* and DNA was extracted from 200 µL of supernatant, eluted in a 50 µL volume. Whole genome sequencing (WGS) was carried out using the MiSeq Illumina pair-end protocol (Illumina, San Diego, CA, USA) and Oxford Nanopore single-long reads (Oxford Nanopore Technologies, Oxford, UK) platforms, as previously described [42,43]. The fastQC command on the Galaxy Europe online platform (https://usegalaxy.eu/, accessed on 1 March 2023) was used to conduct quality control of NGS data. Reads were mapped against the *P. piscolens* reference genome (NCBI accession number: GCF_022846135.1) using CLC software (version 3.15.4, CLC Genomics Workbench QIAGEN Bioinformatics). Mapped reads were assembled to obtain *P. piscolens* genomes. For *M. massiliense* genomes, Illumina and Nanopore reads not mapped on the *P. piscolens* reference genome were de novo assembled using Spades software (Version 3.15.4), and the generated contigs were checked by Blast against the NCBI GenBank database. The quality of genome assemblies was checked using CheckM [44]. Whole genome comparisons were made using PYANI software version (0.2.7) with standard parameters, genome sequences with >95% identity corresponded to the same species [45] (Appendix A). The TYGS (Type (Strain) Genome Server) online tool [46] was used to calculate dDDH (DNA-DNA hybridization) values based on the 25 genome sequences included. The threshold for DDH values indicating membership in the same species was above 70% [47]. The obtained genomes were annotated on the DDBJ Fast Annotation and Submission Tool online platform (https://dfast.ddbj.nig.ac.jp/, accessed on 1 March 2023) (Appendix A), as well as *M. oralis* and *P. piscolens* reference genomes (accession numbers GCF_912073625.1 and GCF_022846135.1, respectively) and *P. piscolens* genomes from this study (accession numbers CAUZXV01000000 and CAUZXU01000000). Phylogenetic analysis based on WGS-derived 16S rRNA gene sequences with the 32 hit blasts was performed using the Neighbor-Join and BioNJ algorithms standard parameters on MEGA software (version 7.0.26) (Appendix A). The Maximum Likelihood method based on the JTT matrix-based model and 1000 replicates bootstrap consensus was used to infer the evolutionary history. Branches corresponding to partitions reproduced in less than 50% of bootstrap replicates were collapsed. The initial tree was automatically obtained by applying the Neighbor-Join and BioNJ algorithms to a matrix of pairwise distances estimated using a JTT model, then selecting the topology with the higher log likelihood value. The final data set comprised 1180 positions.

## 3. Results

### 3.1. Molecular Identification

RT-PCR detection of methanogens was positive in 11 of the 22 (50%) dental pulp samples collected from carious teeth, but in none of the ten wisdom teeth used as negative controls. Sequencing identified *M. oralis* in 7/11 positives (64%), *M. smithii* in one positive (1/11, 9%), and *M. massiliense* in three samples (3/11, 27%). No. 14, No. 73 and No. 74 first hit blasts, respectively, yielded 98% coverage and 98.78% identity (sample No. 14), 99% coverage and 96.9% identity (sample No. 73), and 99% coverage and 98.29% identity (sample No. 74) with *M. massiliense* (Appendix A).

### 3.2. Co-Culture of M. massiliense with P. piscolens

*M. massiliense* was successfully cultured from dental pulp samples No. 14 and No. 74 but not from sample No. 73 or negative controls, using the SAB medium as well as the GG medium under nitrogen atmosphere, with methane being increasingly detected from day 0 to day 20 (Appendix A). Subculture in GG medium was achieved in a minimum of three weeks when cultures turned black with black deposits (Figure 1) and methane was detected. In contrast, culturing *P. piscolens* in GG medium under a nitrogen atmosphere yielded no methane, no hydrogen was detected, and no black deposits were observed (Appendix A). In parallel, subculturing GG broth inoculated with samples No. 14 and No. 74 on COS yielded colonies within days, with MALDI-TOF identification of *P. piscolens* (scores > 2). *P. piscolens* was the only bacterium identified at days 3, 5, 7, 10, and 14. Therefore, two *P. piscolens* isolates were deposited in the CSUR under references Q8451 (from sample No. 14) and Q8452 (from sample No. 74) (Table 1). In addition, although the five *P. piscolens* isolates (including Q8451 and Q8452 here reported) were susceptible in vitro to tetracycline on COS medium (MIC = 20 µg/mL) and in GG medium (50 mg/L < MIC < 100 mg/L), nevertheless attempts to isolate *M. massiliense* in pure culture by incorporation of tetracycline proved unsuccessful under all tested conditions (Appendix A). Finally, two co-cultures of *M. massiliense* and *P. piscolens* from dental pulp samples No. 14 and No. 74 were added to our international public collection CSUR WDCM 875 (Collection Souches Unité des Rickettsies, WDCM 875, Marseille, France) under the references Q8282 and Q8283, respectively.

### 3.3. Microscopy

FISH detected coccobacilli archaea and bacteria in the *M. massiliense*-positive dental pulp sample No. 74, while bacteria but not archaea were detected in the methanogen-negative sample, and the DPBS 1× water negative control yielded no fluorescence (Figure 2). Confocal microscopy observations of *M. massiliense*-*P. piscolens* Q8282 and Q8283 co-cultures revealed autofluorescent 1–2-µm-long and 0.5–0.8-µm-wide coccobacilli corresponding to *M. massiliense*, but no fluorescent coccobacilli of the same size at 420 nm (Figure 1). Furthermore, no *P. piscolens* strain was autofluorescent and no autofluorescence was observed in the negative control (Appendix A). Further electron microscopy observation of Q8282 and Q8283 co-cultures and of *P. piscolens* cultured either in liquid or solid medium showed coccobacilli (0.5–0.8 µm wide and 1–2 µm long) often featuring diplo-coccobacilli (Figure 1 and Appendix A). Therefore, morphological observations alone were not sufficiently discriminative to definitively identify coccobacilli as *M. massiliense* or *P. piscolens*. However, we observed dark crystal structures probably corresponding to the previously observed black deposits in both Q8282 and Q8283 co-cultures and not in *P. piscolens* cultures (Figure 1 and Appendix A).

### 3.4. MALDI-TOF Mass Spectrometry

No interpretable MALDI-TOF spectra were obtained from negative controls, while *E. coli* (positive control) was correctly identified with an identification score of 2.57. Five *P. piscolens* strains cultivated on COS medium were identified with identification scores > 2 and yielded MALDI-TOF-MS spectra identical to the ones derived from culture in liquid GG medium culture, except for one peak which was lacking (average *m*/*z* = 8074.28 *m*/*z* +/−1.64; average relative intensity = 50.68 arbitrary unity (a.u.) +/−14.29). Further, Q8282 and Q8283 co-cultures yielded identical MALDI-TOF-MS spectra, which were superimposable with those derived from *P. piscolens* cultured on COS and GG broth media, except for two additional peaks (average *m*/*z* = 6883.35 *m*/*z* +/−1.04; average relative intensity = 93.2 a.u. +/−5.38 and average *m*/*z* = 6946.40 *m*/*z* +/−1.03; average relative intensity = 79.14 a.u. +/−20.41) and the missing peak reported above, suggesting that these two additional peaks were specific to *M. massiliense* presence in the co-culture (Figure 3).

### 3.5. Whole Genome Sequencing and Analysis

Mapping llumina and Nanopore data against a *P. piscolens* reference genome made it possible to reconstruct two *P. piscolens* genomes derived from Q8282 (Q8451, accession number CAUZXV01000000) and Q8283 (Q8452, accession number CAUZXU01000000) co-cultures. Both genomes are complete without contamination according to CheckM and share similar characteristics together and with the *P. piscolens* reference genome in terms of length, GC% and coding ratio (Table 1). Then, combined assembly of Illumina and Nanopore data yielded two *M. massiliense* genomes derived from Q8282 (GenBank accession number CAUJAM01000000) and Q8283 (GenBank accession number CAUJAN01000000) co-cultures. The former *M. massiliense* genome, here chosen as the reference genome, consisted of a gapless 1,834,388 base pair (bp) molecule displaying a 31.3% GC content. The completeness of this genome, as determined by CheckM, was 100%, without contamination. Genome annotation indicated an 85.9% coding ratio and 1903 protein-encoding genes, 28 tRNA-encoding genes, two rRNA genes, and three CRISPRs (Table 1, Figure 4, Appendix A).

The 16S rRNA gene sequence extracted from WGS exhibited a 99.66% sequence similarity (100% coverage) with the *M. massiliense* Q8283 genome, 99.84% similarity (85% coverage) with *Methanobrevibacter* sp. N13 (*M. massiliense*), 98.44% sequence similarity (100% coverage) with *Methanobrevibacter* sp. YE315, a methanogen isolated from pooled rumen of zebus (*Bos indicus*), 97.85% (91% coverage) with *M. smithii*, 97.82% (90% coverage) with *Methanobrevibacter gottschalkii* (*M. gottschalkii*) isolated from horse and pig faeces [48], 97.47% (85% coverage) with *Methanobrevibacter thaueri* (*M. thaueri*) isolated from cow faeces [48] and 97.09% (99% coverage) with *M. oralis* (Appendix A). Accordingly, the 16S rRNA gene sequence-derived phylogenetic tree clustered *M. massiliense* Q8282 and Q8283 with *Methanobrevibacter* sp. N13, then with *Methanobrevibacter* sp. YE315, *M. thaueri* and *M. gottschalkii*, and *Methanobrevibacter millerae* (*M. millerae*) (Figure 5).

Average Nucleotide Identity (ANI), including 15 genomes, indicated that the two *M. massiliense* genomes share 99% identity but were distant from any other cultured *Methanobrevibacter* species. Indeed, the reference genome of *M. massiliense* displayed an ANI of 80% with *M. gottschalkii*, followed by *Methanobrevibacter* sp. YE315 (ANI, 79%), *M. thaueri* (ANI, 78%), and *M. oralis* (ANI, 78%) (Figure 6). The dDDH values of the two genomes with other methanogen species confirmed these previous results (Appendix A). The *M. massiliense* genome annotation uncovered features absent from the *M. oralis* reference genome, including malate dehydrogenase (encoded by LOCUS_01990 and LOCUS_13560), molybdate permease (*modB*, encoded by LOCUS_01740 and LOCUS_19870), and nicotianamine synthase (encoded by LOCUS_00800 and LOCUS_19110). Additionally, both *P. piscolens* genomes (LOCUS_18390, LOCUS_05780, and LOCUS_15000) and *M. massiliense* genomes (LOCUS_01130 and LOCUS_18360), contained pyridoxamine kinase involved in vitamin B6 synthesis, while *M. oralis* possessed PDX genes (LOCUS_15160 and LOCUS_18540). Furthermore, *P. piscolens* genomes exhibited various metal ion dehydrogenases.

## 4. Discussion

In this study, methanogens were identified in half of the pulp samples extracted from carious teeth, consistent with findings from a prior study [24]. *M. oralis* emerged as the predominant species, followed by *M. massiliense* and *M. smithii*. This outcome aligns with some observations in periodontitis and peri-implantitis within dental plaque [16,17]. However, in contrast to earlier reports of methanogens in saliva and dental plaque from healthy individuals and around healthy implants [11,17,18], no methanogens were detected in healthy dental pulp. This absence is unsurprising, given the enclosed nature of the pulp chamber, which is secluded from the oral cavity. This observation leads to the hypothesis that methanogens may colonise microorganisms involved in the endodontic infection process. Notably, *M. massiliense* was the sole methanogen cultured from endodontic sites, leading to two *M. massiliense*–*P. piscolens* co-cultures obtained by culturing necrotic pulp samples collected from carious teeth in two different patients. The comprehensive characterisation of this fastidious methanogen was achieved through various techniques, including microscopy, genomic analysis, and phylogenetic investigation, bolstered by the implementation of negative controls to ensure data validity. Both co-cultures successfully subcultured in GG medium [34] were deposited in the public collection CSUR, and their genomes were made available in the GenBank public database.

Interestingly, we detected the presence of the same associated bacterium, *P. piscolens*, in two separate individuals, which aligns with previous findings in a case of severe periodontitis [28]. *P. piscolens* belongs to the lesser-explored phylum Synergistetes and is a Gram-negative coccobacillus originally isolated from oral samples [40]. While the molecular detection of *P. piscolens* in endodontic samples was previously established, our culture-based approach provides additional evidence of its potential involvement in endodontic infections. *P. piscolens* has also been identified in diverse human specimens, including the gut [49,50], stool samples, and the sinuses [51]. Furthermore, it has been detected in animals, such as the gut microbiota of pigs [52], cattle rumen [53], and subgingival microbiota of horses [54]. Investigating the presence of *M. massiliense* in animals could shed light on new oral and also non-oral ecological niches for this methanogen, potentially uncovering zoonotic sources and transmission pathways.

Notably, *M. massiliense* appeared to be more challenging to cultivate than other methanogens, as we were unable to eliminate *P. piscolens* without affecting the viability of *M. massiliense*, indicating a strong dependency of methanogen on its partner. Despite attempts to enhance *M. massiliense* growth by increasing acetate and fatty volatile acids typically produced by *P. piscolens* [40], no significant improvement was observed. However, our hypothesis is that hydrogen may not be required for *M. massiliense* growth, as it grew in GG medium under a nitrogen atmosphere, and as *P. piscolens* did not produce hydrogen in our cultures in GG medium, as previously described [40]. Genome annotation also revealed the presence of a formate dehydrogenase, confirming this possibility.

The association between *M. massiliense* and *P. piscolens* in a co-culture revealed a transkingdom dependency (Archaea and Bacteria). Interestingly, while *P. piscolens* was isolated independently, *M. massiliense* was not, leading us to hypothesise that *M. massiliense* is reliant on *P. piscolens* for sustenance, supporting a unidirectional dependency. *M. massiliense* possesses a malate dehydrogenase that is not present in *M. oralis,* which has been isolated in pure culture [55]. Malate may play a crucial role, potentially serving as a source for acetyl-CoA, which is essential for various metabolic reactions, including the biosynthesis of the coenzyme methylcobalamin (Methyl-COB), an important cofactor in methanogenesis [56]. *P. piscolens* may have the capacity to produce malate through fumarate hydratase, suggesting a cooperative relationship, where one provides the required malate for important metabolic reactions in the other. However, fumarate hydratase is widespread in bacteria, and its presence in *P. piscolens* may not be sufficient to explain the uniqueness of *M. massiliense* dependency. However, the high metal requirement of *M. massiliense*, evident as it possesses nicotianamine synthase and a molybdate transporter absent in *M. oralis*, raises intriguing questions. Particularly, nicotianamine is a type of metallophore, a small organic molecule used to chelate metals, aiding in their absorption and transport within cells [57]. Dark crystal structures observed in co-culture may corroborate these findings. Metal ions are crucial co-factors in various metabolic pathways, including methanogenesis [58], and *P. piscolens* harbours several metal-ion-dependent dehydrogenases, possibly involved in electron transfer between the two species. Both share the same vitamin B6 biosynthesis pathway through pyridoxal kinase, while *M. oralis* employs an alternative pathway via PDX1 and PDX2 [10]. This suggests shared essential co-factors for diverse metabolic functions between *M. massiliense* and *P. piscolens*.

This intricate association underscores the importance of cooperation between different species, spanning domains, in microbial ecosystems. Sharing essential metabolites and co-factors enables these organisms to thrive together but also highlights the importance of studying metabolic pathways, testing substrate affinity, and making systematic efforts to enhance *M. massiliense* culture. Investigating the production and consumption of metabolites over time has become crucial to understanding how they interact. This knowledge can be valuable in finding ways to obtain a pure culture of *M. massiliense*. While the precise mechanisms of this association remain unresolved, delving into these interactions may provide valuable insights into their roles in human health and disease. Understanding these microbial relationships not only advances our fundamental knowledge of the microbiome but also opens avenues for potential therapeutic interventions and preventive strategies in oral health, promising advancements in both oral and systemic health landscapes.

## 5. Conclusions

In conclusion, our study not only reinforces the potential involvement of methanogens in endodontic infections, as indicated by an observed prevalence of 50% in infected pulps, but also emphasises the rediscovery of *M. massiliense* in a novel pathological context. Notably, *M. massiliense* emerges as the second prevalent methanogen, coexisting with *P. piscolens.* This unique association demonstrates a one-way dependency, where *M. massiliense* relies on *P. piscolens* for growth and survival. Genomic analysis highlighted possible metabolic cooperation between these species. Nevertheless, further research, encompassing in-depth exploration of metabolic pathways, production and consumption of metabolites, extensive large-scale testing of substrate affinity, and persistent isolation efforts, is imperative for a more comprehensive understanding. This intricate dependence underscores the importance of cooperation in microbial ecosystems. Gaining insights into these interactions would be useful for unravelling the complexities of human health and disease. Although the exact mechanisms require further investigations, this study opens avenues for future research on possible therapeutic interventions and preventive strategies in oral health, promising improvements in both oral and overall health.

## Figures and Tables

**Figure 1 microorganisms-12-00215-f001:**
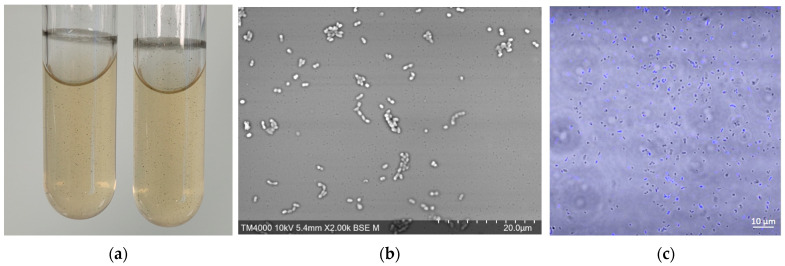
Macroscopic and microscopic features of *Methanobrevibacter massiliense*. (**a**) Macroscopic observation of *M. massiliense* and *P. piscolens* co-cultures, dark deposits are visible. (**b**) Electron microscopy (TM4000 HITACHI, 10 KV, × 1.500), coccobacilli are visible, but no morphologic features made it possible to distinguish between *M. massiliense* and *P. piscolens*. (**c**) Confocal microscopy merged view of autofluorescence at 420 nm and brightfill mode; autofluorescent coccobacilli are visible as well as non-autofluorescent coccobacilli (LSM 900 (Carl Zeiss Microscopy GmbH)).

**Figure 2 microorganisms-12-00215-f002:**
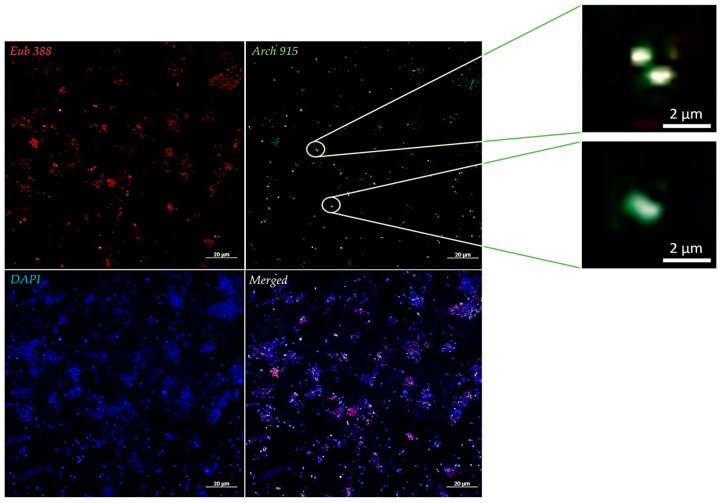
Fluorescence in situ hybridization (FISH) representative detection of *M. massiliense* and bacteria in a direct necrotic pulp sample. Eub 388 probe staining bacteria 16S rRNA gene in red. Arch 915 probe staining the archaeal 16S rRNA gene in green. Universal DNA DAPI staining in blue. Merged. Scale bar, 20 μm.

**Figure 3 microorganisms-12-00215-f003:**
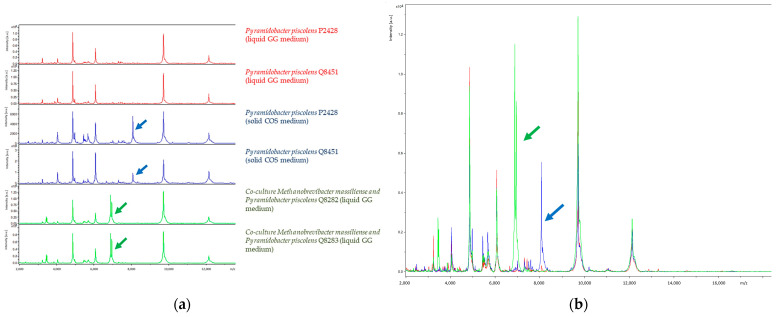
MALDI-TOF spectra and superposition of spectra from co-cultures Q8282 and Q8283 with spectra from *P. piscolens* cultivated on COS and in liquid GG medium. (**a**) MALDI-TOF spectra of *P. piscolens* cultivated on COS (in red), MALDI-TOF spectra of *P. piscolens* cultivated in liquid GG medium (in blue), MALDI-TOF spectra of co-cultures Q8282 and Q8283 (in green), smoothed and baseline substrated. (**b**) Superposition of the six spectra. All the spectra are superimposable, with one missing peak in the co-culture of *M. massiliense* and *P. piscolens*, the same as that absent in the liquid medium spectra (blue arrow), and two additional peaks in the co-cultures with *M. massiliense* (green arrow). The superposition of the spectra reveals a significant correlation between co-cultures Q8282 and Q8283 with strains of *P. piscolens* cultivated on COS, as well as in liquid GG medium. These observations suggest a specific spectrum associated with *M. massiliense* in the co-cultures.

**Figure 4 microorganisms-12-00215-f004:**
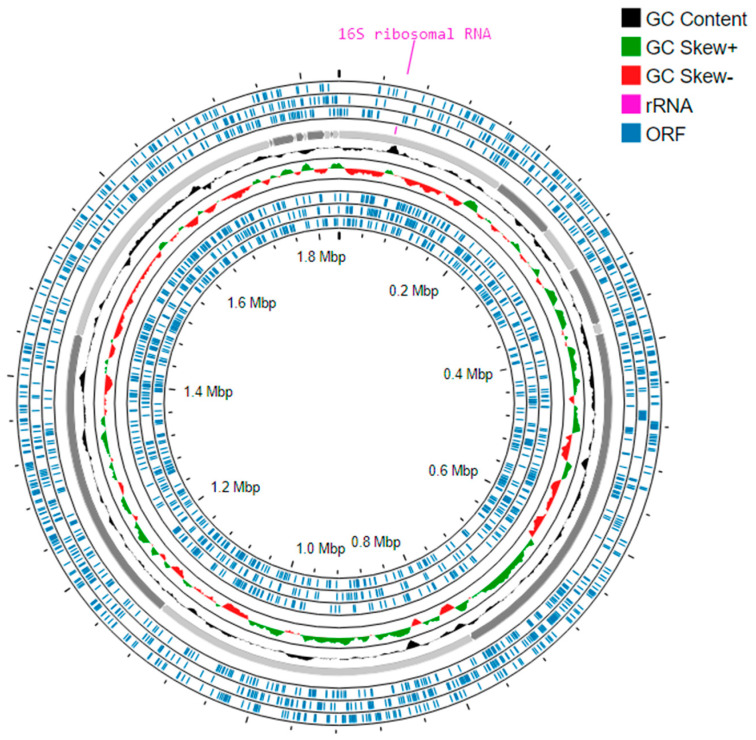
Circular representation of the *Methanobrevibacter massiliense* reference genome. The genome was visualised using the Proksee platform with default settings (version 1.0.0), available at https://proksee.ca/, accessed on 1 March 2023. The 19 contigs are displayed with colour-coded highlights, allowing for the identification of key genomic elements, and providing insight into the composition and structure of the *M. massiliense* genome. Open-reading frames are shown in blue, GC content is represented in black, while positive and negative GC skew are depicted in green and red, respectively. The rRNA regions are highlighted in pink. This analysis provides an overview of the distribution of protein-coding genes and reveals variations in GC content throughout the genome.

**Figure 5 microorganisms-12-00215-f005:**
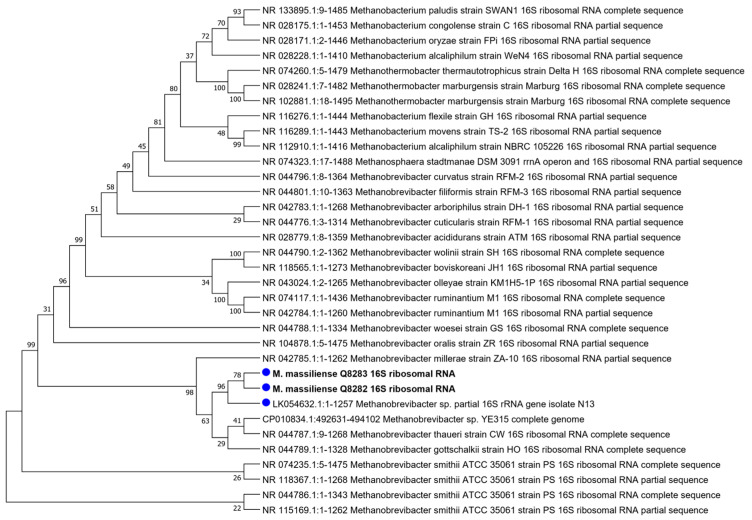
Maximum Likelihood phylogenetic tree based on 16S rRNA sequence analysis of *M. massiliense* and the first 35 hit blasts downloaded from NCBI GenBank Database (3 January 2023).

**Figure 6 microorganisms-12-00215-f006:**
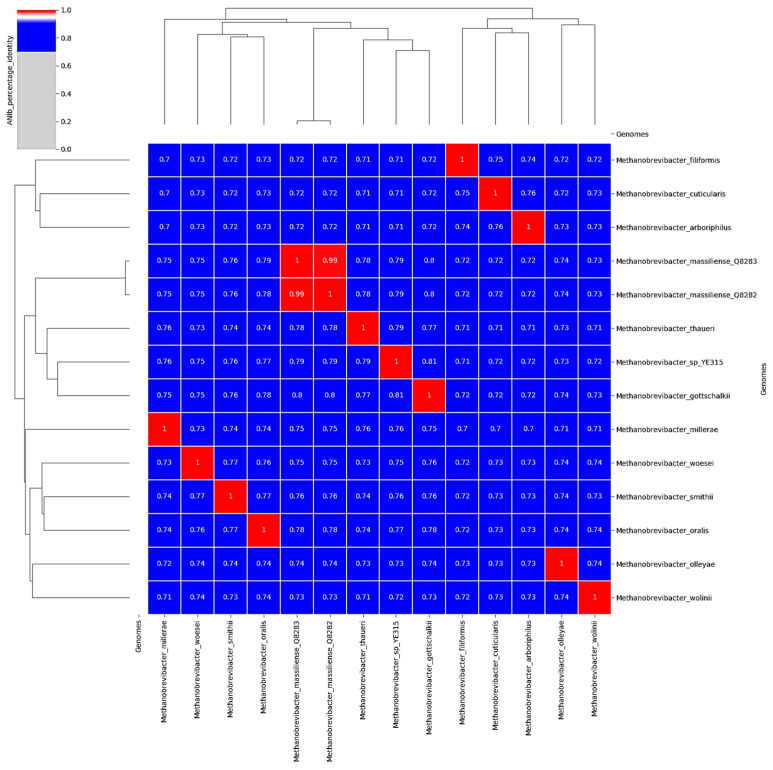
Heat map phylogeny generated with ANI values using PYANI software version (0.2.7) with standard parameters of *M. massiliense* and closely related methanogen species.

**Table 1 microorganisms-12-00215-t001:** Characteristics of *Methanobrevibacter massiliense* and *Pyramidobacter piscolens* genomes retrieved in this study.

	*M. massiliense* Q8282	*M. massiliense* Q8283	*P. piscolens* Q8451	*P. piscolens* Q8452
Total sequence length (bp)	1,834,388	2,299,297	2,662,074	2,638,206
Number of sequences	19	15	11	4
Longest sequences (bp)	384,479	551,700	846,238	1,351,082
N50 (bp)	325,131	400,054	616,914	1,351,082
Gap Ratio (%)	0	0	0.001503	0.078728
GC content (%)	31.3	30.6	59.6	59.8
Number of CDSs	1903	2332	2512	2438
Average protein length	275.8	277.9	310.9	315.8
Coding ratio (%)	85.9	84.5	88	87.5
Number of rRNAs	2	4	5	12
Number of tRNAs	28	32	55	57
Number of CRISPRs	3	2	2	3
CheckM completeness (%)	100	97.67	100	100
CheckM contamination (%)	0	4.72	0	0

## Data Availability

All genomic data were submitted to the GenBank NCBI database under accession numbers CAUZXV01000000, CAUZXU01000000, CAUJAM01000000 and CAUJAN01000000.

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
