# Peer review of "Methanobrevibacter massiliense and Pyramidobacter piscolens Co-Culture Illustrates Transkingdom Symbiosis"

_microorganisms, 2024, doi:10.3390/microorganisms12010215_

Round 1

Reviewer 1 Report

Comments and Suggestions for Authors

I suggest to extend the introduction. The current version describes the microorganism then studied in the article; however, it does not introduce the specificity of the study. In this sense, a brief state of the art, discussing previous findings and also pionieristic targets, should be added to this section.

The result showed in Section 3.1 is already known; in fact, it was previously mentioned in the Introduction. Here the authors should insert a discussion about, besed on their results, their expertise and previous studies, methanogens were collected exclusively from carious teeth.

Based on the results achieved in the present study, can the authors explain in what the future studies aill consist of?

Comments on the Quality of English Language

The quality of english language is suitable for publication.

Reviewer 2 Report

Comments and Suggestions for Authors

In this study, the authors report the rediscovery of M. massiliense  in a new pathological context  through cultures from necrotic dental pulp samples. They present new data, including genomic and microscopic features, which contribute to a more comprehensive characterization of this challenging methanogen. A total of thirty-two molars and premolars were collected from 32 patients as part of 66 routine practice by dentists and dental students under the supervision of a graduate dentist at the Pôle Odontologie Timone (Assistance Publique des Hôpitaux de Marseille) from 68 September 2022 to February 2023. Twenty-two teeth had to be extracted for carious reasons and ten teeth were extracted as wisdom teeth; none were in the state of root debris  or fractured during the extraction. Two M. massiliense-P. piscolens co-cultures obtained by culturing necrotic pulp samples collected from carious teeth in two different patients enabled a comprehensive characterization of this fastidious methanogen through various approaches, including microscopy, genomic analysis, and phylogenetic investigation in the presence of negative controls validating data. Both co-cultures successfully subcultured in GG medium  were deposited in the public collection CSUR, and their genomes were made available in the GenBank public database. In conclusion, the study presents the rediscovery of M. massiliense in endodontic infection, coexisting with P. piscolens. This unique association demonstrates a one-way dependency, where M. massiliense relies on P. piscolens for growth and survival. Genomic  analysis highlighted possible metabolic cooperation between these species. This intricate dependence underscores the importance of cooperation in microbial ecosystems. Understanding these interactions would provide valuable insights into human health and disease. Although exact mechanisms require further investigations, this study opens avenues for future research on possible therapeutic interventions and preventive strategies in oral health, promising improvements in both oral and overall health. 

The paper is well done but I have some remarks:

- Moderate editing of English language required

- The figure are not clear (figures 2, 3, 6)

- The conclusion have to expanded

Comments on the Quality of English Language

Moderate editing of English language required
